# Designing a Water-Immersed Rectangular Horn Antenna for Generating Underwater OAM Waves

**Yang Yang** [1], **Zhanliang Wang** [1], **Shaomeng Wang** [1], **Qing Zhou** [2], **Fei Shen** [3], **Haibo Jiang** [4], **Zhe Wu** [1], **Baoqing Zeng** [1], **Zhongyi Guo** [3,5,*] and **Yubin Gong** [1,*]

[1]  National Key Lab on Vacuum Electronics, University of Electronic Science and Technology of China, Chengdu 610054, China; yangyang_19930129@163.com (Y.Y.); wangzl@uestc.edu.cn (Z.W.); wangsm@ntu.edu.sg (S.W.); zhewu@uestc.edu.cn (Z.W.); bqzeng@uestc.edu.cn (B.Z.)
[2]  Hebei Key Laboratory of Compact Fusion, Langfang 056000, China; ZQ901126@126.com
[3]  School of Electrical Engineering & Intelligentization, Dongguan University of Technology, Dongguan 523808, China; shenfei@hfut.edu.cn
[4]  Chengdu Institute of Biology, Chinese Academy of Sciences, Chengdu 610041, China; jianghb@cib.ac.cn
[5]  School of Computer and Information, Hefei University of Technology, Hefei 230009, China
*  Correspondence: guozhongyi@hfut.edu.cn (Z.G.); ybgong@uestc.edu.cn (Y.G.); Tel.: +86-186-5515-1981 (Z.G.); +86-138-0803-6055 (Y.G.)

**Abstract:** In order to extend the applications of vortex waves, we propose a water-immersed rectangular horn antenna array for generating underwater vortex waves carrying the orbital angular momentum (OAM). Firstly, a single dielectric-loaded rectangular horn antenna with the central frequency of 2.6 GHz was designed for generating underwater electromagnetic (EM) waves. Due to the supplementing dielectric-loaded waveguide in this single antenna, the problems with difficult sealing and fixation of the feed probe could be solved effectively. The simulation results show that it has a good impedance characteristics ($S_{11} < -10$ dB) and reasonable losses (less than 3.5 dB total for two antennas and a coaxial line) from 2.5 GHz to 2.7 GHz. Experiments on the single antenna were also carried out, which agree well with the simulations. Based on the designed single antenna, the water-immersed rectangular horn antenna array was proposed, and the phase gradient from $0\sim 2\pi$ was fed to the horn antennas for generating underwater OAM waves. The simulation results demonstrate high fidelity of the generated OAM waves from the intensity and phase distributions. The purity of the generated OAM modes was also investigated and further verifies the high fidelity of the generated OAM waves. The generated high-quality OAM waves meet the requirements for underwater applications of OAM, such as underwater communication and underwater imaging.

**Keywords:** water-immersed; orbital angular momentum (OAM); rectangular horn antenna array; dielectric-loaded

## 1. Introduction

As is well known, vortex waves carrying orbital angular momentum (OAM) have a rotating phase factor of $e^{il\theta}$, where $\theta$ is the azimuth angle, and $l$ is an integer number called "topological charge", which corresponds to the order of the OAM modes. Since the quantum characteristics of the OAM states were first discovered by Allen [1] in 1992, the OAM-carrying beam has attracted lots of attention. Then, in 2007, a uniform circular array with a successive phase shift was proposed to generate OAM beams in the radio frequency domain [2]. After that, studies of OAM were introduced into the radio frequency domain. In fact, OAM can provide rotational degrees of freedom to the electromagnetic (EM) field, which would be advantageous for manipulating particles [3–5],

quantum information processing [6], communication systems [7–10], and imaging systems [11–13]. Especially in communication and imaging systems, the different orthogonal OAM states can span a Hilbert space of infinite dimensions, providing a novel degree of freedom to increase communication capacity and to improve imaging resolution, respectively.

In order to utilize the OAM in the radio domain, one should first generate the vortex waves carrying OAM. There are numerous generation methods of OAM-carrying waves in the radio domain [14]: antenna array [8,15,16], microstrip patch antenna [17,18], spiral phase plate (SPP) [19], spiral antennas [20–22], slot antennas [23], cylinder dielectric resonator antennas [24], horn lens antennas [25], and so on. Unfortunately, due to the impedance mismatch between the antenna and coaxial line of 50 Ω, the underwater vortex waves cannot be generated by all of these structures efficiently, which prohibits the underwater applications of OAM, such as underwater communication and underwater imaging. Recently, two types of double-ridge horn antennas [26,27] have been reported to generate underwater EM waves. With the water-filled coaxial line impedance structure, the double-ridge horn antenna in [26] can generate underwater EM waves while the horn antenna in [27] realizes the same function by designing a double-ridge curve. However, there exists a problem of difficult sealing in both antennas, resulting in the short circuit of the coaxial cable, so that it cannot emit EM waves effectively. Moreover, horn [28], several resonant [29,30] and travelling wave [31]-type antennas have been proposed for breast imaging systems, but they cannot all work well in water. Therefore, it is necessary to investigate an alternative approach with a better seal for generating underwater EM waves efficiently.

In this paper we describe the design and testing of a single water-immersed rectangular horn antenna, which can generate underwater EM waves. By introducing a dielectric-loaded waveguide in this antenna, the sealing problem of the antenna can be solved, while the established position and length of the feed probe can be ensured, so the frequency offset can be reduced to a certain extent. The experimental results of a single antenna are consistent with the simulation ones. Based on the above single antenna, we designed and simulated a water-immersed OAM rectangular horn antenna array. Through simulations, the high-fidelity vortex fields with the OAM modes of $l = 0, 1, 2, 3$ can be generated efficiently in a far field, and meanwhile, we obtained the purity of these OAM modes from the vortex field distributions. The whole paper is organized as follows: Firstly, the reasons for the proposed antenna and antenna array are introduced. Then, the single antenna is designed, and simulated, and the measured results are demonstrated in the second section. Based on the single antenna, the OAM-generating antenna array is simulated in the third section. Finally, some useful conclusions are provided.

## 2. Single Antenna

In this section, we discuss the design of a dielectric-loaded single horn antenna for generating underwater EM waves, and its simulation is carried out with a computer simulation technology (CST) microwave studio [32]. In order to calculate the loss of the antenna in deionized water, a box filled with deionized water was selected to replace the air box, while the background medium was set as the water. The open boundaries (PML) were set to the six faces of the water box to simulate the infinite space. Moreover, the waveguide port was used to feed the antenna at the bottom of the coaxial line.

The structure of the designed single antenna is presented in Figure 1. The waveguide of the horn antenna was filled with a higher relative dielectric constant ceramic medium to solve the sealing problem under the premise of ensuring impedance matching. As is well known, the ceramics medium is hard and crumbly, so it cannot be used in a patch antenna. Of course, the dielectric-loaded parabolic antenna has a larger structure, which is not suitable for the relevant applications of underwater EM waves. Moreover, across the frequency range of 2.5–2.7 GHz, the relative dielectric constant of $Z_rO_2$ is almost equal to 36, while the loss tangent is a low constant, from which the medium $Z_rO_2$ can be regarded as a non-dispersive material for impedance matching but with lower loss.

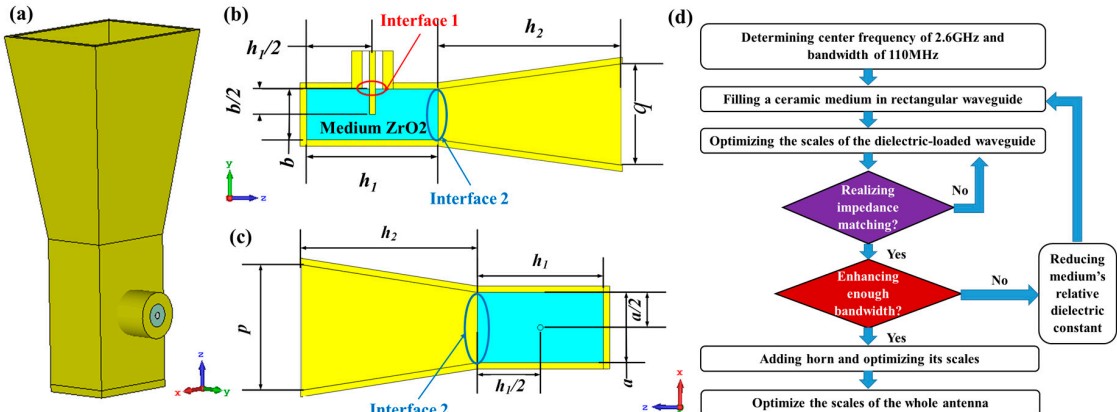

**Figure 1.** The structure of the single designed horn antenna; (**a**) three dimensional view, (**b**) two dimensional view of the YOZ plane, (**c**) two dimensional view of the XOZ plane, and (**d**) the step-by-step design procedure.

The medium facilitates the coaxial sealing at interface 1 between the coaxial line and the waveguide, as shown in the red circle in Figure 1b. With the $Z_rO_2$ medium filling in the waveguide, only the sealing at interface 2 (blue circle in Figure 1b,c) between the horn and the waveguide is necessary, which can be easily realized compared to that at interface 1. In addition, the origin is only selected at the center of interface 2. Another benefit of the zirconium oxide ($Z_rO_2$) medium is that the established position and length of the probe can be ensured, so the frequency offset can be reduced to a certain extent.

Furthermore, the impedance calculation of the rectangular waveguide with coaxial excitation should strictly follow the reported method [33], and a step-by-step design procedure is provided in Figure 1d. Based on the reported conclusions [28], in order to increase the bandwidth, the relative dielectric constant of the ceramic medium should be reduced, but the aperture of the antenna should be increased, which hinders the miniaturization and related applications of the antenna. Therefore, the relative dielectric constant of the ceramic medium should be reduced in a certain range. Due to the change of the relative dielectric constant of the packed medium, the dimensions of the rectangular waveguide and feed probe need to be optimized, and the values of the optimized parameters are listed in Table 1. With these optimized parameters, the radiation pattern of the designed antenna will keep stable and optimized as well. In addition, a horn is added to the dielectric-loaded waveguide to obtain higher gain and better directivity. Therefore, although the loss of water is larger, the radiation signal from the single antenna can transmit for a longer distance in water, which provides enough distance for the superposition of the radiation signal to produce underwater OAM waves.

**Table 1.** The main parameters and the corresponding values of the single antenna.

| Parameters | Values (mm) |
|:---:|:---:|
| $h_1$ | 20 |
| $h_2$ | 28 |
| a | 11.15 |
| b | 8 |
| p | 20 |
| q | 16 |

A coaxial line with a characteristic impedance of 50 Ω was placed perpendicular to the rectangular waveguide for exciting the transverse electric zero one ($TE_{01}$) mode microwave signal. The length of the probe equals the radius of the waveguide to make sure the radiation field is located at the center area. The rectangular horn antenna was selected due to the following advantages (compared with a circular horn antenna):

(i) Fewer merger patterns in the rectangular waveguide than that in the circular waveguide: there are fewer merger modes in the rectangular waveguide compared with that in circular waveguide.

(ii) Simplicity of processing: the medium $Z_rO_2$ is a kind of ceramic and it is hard and crumbly, so, compared to a circular shape, the rectangle is much easier to fabricate.

(iii) Reliability of antenna array assembly: due to the fact that it is easier to identify the direction of the rectangular horn antenna than that of the circular one, it is more reliable to generate OAM waves by the rectangular horn antenna array.

To measure $S_{11}$ of the single antenna and $S_{21}$ between the two antennas, a set of swept transmission loss measurements were implemented. As shown in Figure 2a, the tested antennas were placed at the center of a 40 cm × 24 cm × 35 cm glass tank at a depth of 15 cm, and the tank was filled with deionized water whose permittivity is 78.4, loss tangent is 0.125 and conductivity is 1.42 S/m respectively at 2.6 GHz. Due to the large loss of water and the adequate space of the tank, noises and most reflected waves can be absorbed, so it can be regarded as an infinite space that is also called an "anechoic environment". Two antennas were fixed by two aluminum alloy supports, and there was an adjustable clamp at the end of the support to fix the antenna. A set of measurements were taken to measure the power patterns of this antenna, as shown in Figure 2b. Due to the large transmission loss of the wave in water, it was necessary to utilize an amplifier with a 20 dB gain for amplifying the received signal. The energy of the radiation signal received by another antenna will decrease as the angle increases. If the angle is large enough, the received signal will be too small to be extracted from the noise. The schematic maps corresponding to the experimental setups in Figure 2a,b are presented in Figure 2c,d, respectively.

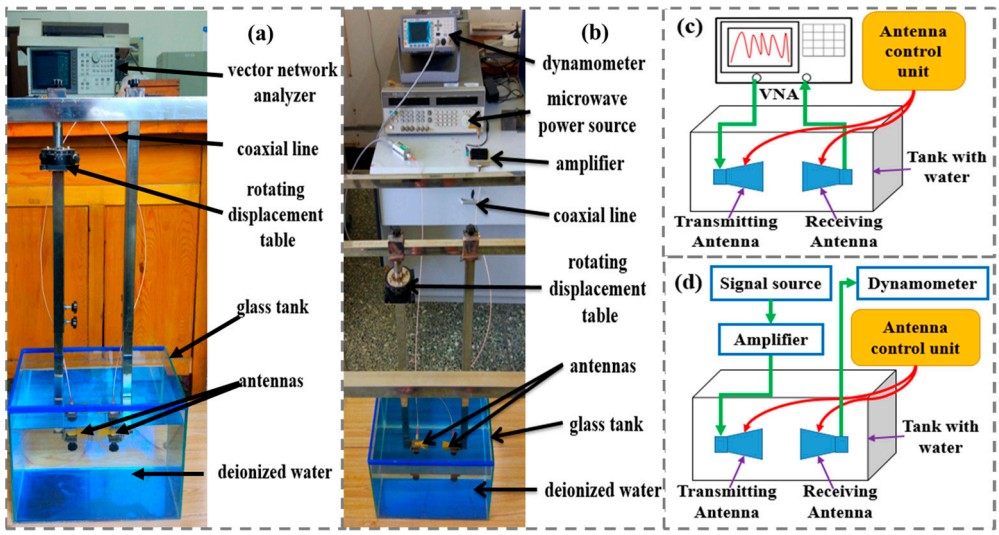

**Figure 2.** The experimental configurations of antenna transmission measurements (**a**) and antenna pattern measurements (**b**). The schematic maps (**c**) and (**d**) correspond to the experimental setups in (**a**) and (**b**).

Figure 3a demonstrates the comparisons of $S_{11}$ of the single antenna and $S_{21}$ of two antennas with a distance of 0.5 cm submerged in deionized water between the experimental and simulated results. The experimental results of $S_{11}$ are lower than the simulation ones due to the different absorbed energies by deionized water and some machining errors. The box in the simulation, when calculating $S_{21}$, was filled with deionized water, which is characterized by the Debye first model. The relative dielectric constants of water from 2.5 GHz to 2.7 GHz in simulation can be found in Figure 3b [34]. In addition, $S_{21}$ depends on the total energy *(P)* of the signal fed into the antenna at port one and the measured energy $S$ of the signal by the receiving antenna at port two. Furthermore, $S$ can be calculated by Equation (1):

$$S \approx P - R - L \tag{1}$$

where $R$ depending on $S_{11}$ represents the total reflection energy of the antenna, and $L$ is the absorption loss by water. Here, $P$ is considered to be the same in the simulation and experiment. $L$ is larger than $R$ due to the large absorption loss of the wave by water and the good impedance characteristics. In other words, $R$ can be ignored when calculating $S$. Therefore, although the experimental and simulated $S_{11}$ do not agree very well, the experimental $S_{21}$ may be very close to the simulation one.

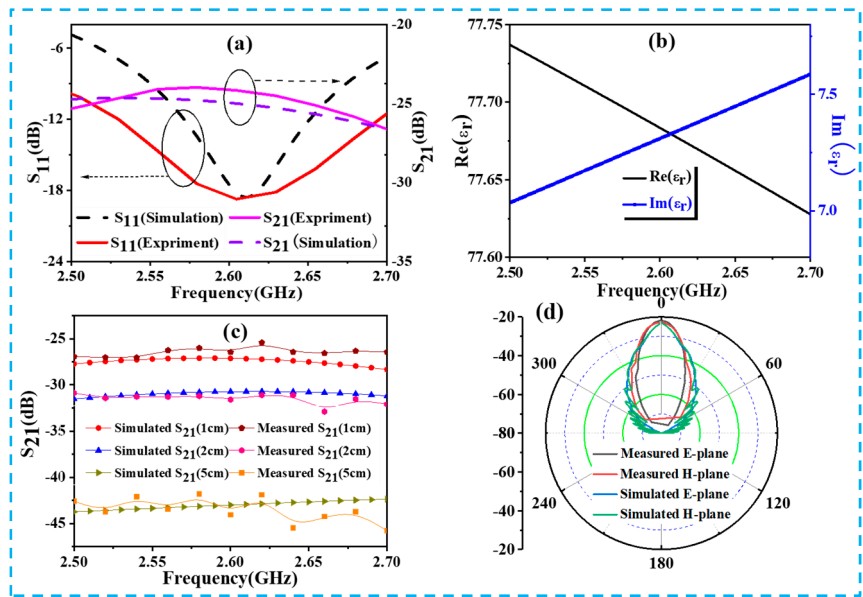

**Figure 3.** (**a**) $S_{11}$ of the single antenna and $S_{21}$ of the two antennas in water; (**b**) the relative dielectric constant of water from 2.5 GHz to 2.7 GHz using first Debye mode; (**c**) simulated and measured $S_{21}$ with different distances between two antennas; (**d**) measured and simulated power patterns in the E and H plane at 2.6 GHz.

$S_{21}$ is about −24.17 dB at 2.6 GHz, and the concrete transmission loss (382 dB/m at 3 GHz) of the wave in water is about 23.3 dB (6.1 cm). Therefore, the combined loss from both antennas and coaxial lines is 0.7 dB at 2.6 GHz. Even at the higher frequency of 2.7 GHz ($S_{21}$ is about −26.8 dB), the total loss was only 3.5 dB, which is still acceptable. Moreover, $S_{21}$, with different distances between the big mouths of the two antennas, is shown in Figure 3c. The loss of water is relatively large, and the water with different conductivities may have larger differences in the loss values. Although there are few differences between simulated and measured data in Figure 3c, they still have the same change trend, which can be another verification of the above conclusion. The value of $S_{21}$ with a distance of 50 mm is higher than −50 dB, which can still be detected by a vector network analyzer (VNA), demonstrating the fact that the antenna can radiate at least 50 mm in water. In addition, the values of $S_{21}$ with the distance of 5 mm and 50 mm between the two antennas were about −25 dB and −45 dB in [26], respectively, which agree well with that of our proposed work, as well as the value of $S_{21}$ (about −35 dB) with the distance of 20 mm between the two antennas in [27].

The 0 dBm input signal from the microwave power source is first amplified 20 dB by the power amplifier, and it then arrives at the transmitting antenna. The radiation field by the transmitting antenna is finally received by another antenna and measured by an Agilent N1913A power meter (Keysight, Santa Rosa, CA, USA). In this process, the combined loss of all the coaxial lines is about 3 dB. By changing the angle of rotating displacement table and adjusting the antenna orientation, the power patterns in the E and H plane at 2.6 GHz with the distance of 5 cm between the big mouths of the two antennas can be obtained, as shown in Figure 3d. The 3 dB angular width in the E plane and H plane are about 13 degrees and 15.5 degrees, respectively, which demonstrates good directionality of this antenna. Subtracting the 20 dB gain of the amplifier, and adding the 3 dB loss of the coaxial line, 33 dB transmission loss (5 cm) and the 40 dB absorption loss of water (10.6 cm), the gain of the antenna is

about 13.5 dB at 0 degrees. The corresponding simulated power patterns are also shown in Figure 3d, which are consistent with the experimental ones to a certain extent, and the small differences may come from the different loss of water in the simulation and experiment, the influence of the experimental devices, and the measured errors.

## 3. Antenna Array

As the approach based on the phased uniform array is very flexible and easily controlled, it is suitable for OAM-generation in water. Based on the above designed single antenna, an OAM antenna array is put forward for generating underwater vortex waves in this section.

The schematic configurations for the OAM-generating system are shown in Figure 4a. The RF signal coming from the VNA is first amplified by an amplifier. Then the amplified signal is transmitted into the eight-way power divider and the eight-way phase shifter to feed the antenna array accordingly. The received signal by the receiving antenna is finally saved on a computer through the VNA. The model of the array is shown in Figure 4b. The N antennas are located equidistantly around the perimeter of the circle and are fed with a phase difference between each element $\delta\varnothing = 2\pi l/N$, where $l$ denotes the OAM modes. The phase shifts of eight antennas with different OAM states are shown in Table 2. The radius of this concentric circle is set as 56 mm, while the center frequency is 2.6 GHz. The single antenna is linearly polarized in the X-direction, so the feed coaxial line of eight elements is placed in the same direction to generate the linear-polarization OAM wave.

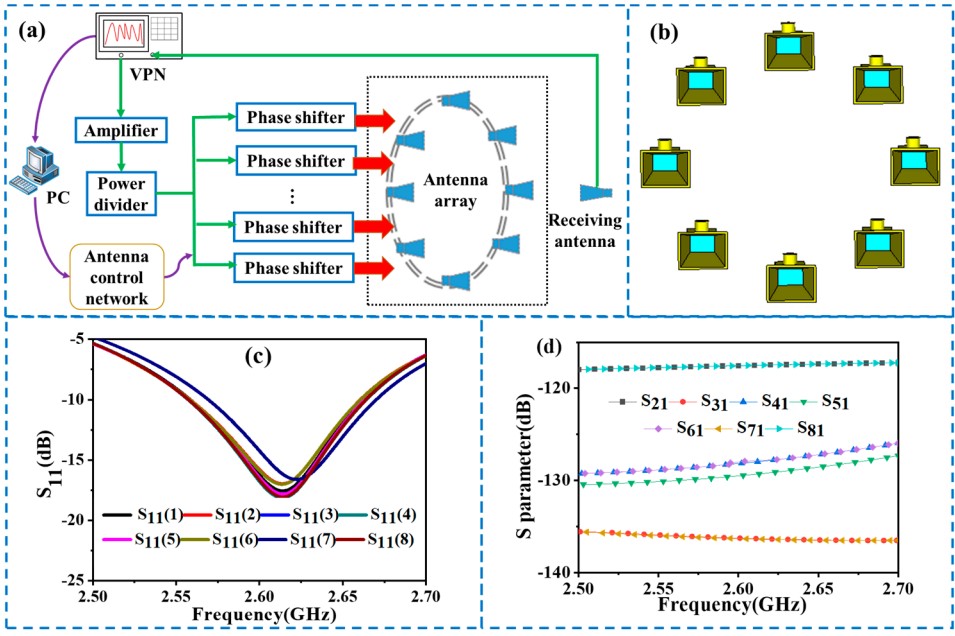

**Figure 4.** (**a**) Schematic configuration for generating OAM beams in water; (**b**) geometry of the OAM horn antenna array; (**c**) $S_{11}$ of eight elements; (**d**) $S$ parameters between the first element and the others.

The $S_{11}$ of eight used antenna elements are shown in Figure 4c, from which we can see that the water-immersed rectangular horn antenna array performs very well in the frequency range of 2.55–2.68 GHz. In addition, the $S_{11}$ of any antenna is almost the same as the others, which means there is good consistency among the antenna array elements. The presented $S$ parameters between the first element and the others, as depicted in Figure 4d, are all below −115 dB, which demonstrates the fact of less coupling among the eight used elements. Therefore, the far field of each element can be superimposed to produce a linearly polarized vortex field.

**Table 2.** The phase shifts of eight antennas with different OAM states.

| Phase Shift (Degree) Antenna | OAM States | | | |
|---|---|---|---|---|
| | $l = 0$ | $l = 1$ | $l = 2$ | $l = 3$ |
| 1 | 0 | 0 | 0 | 0 |
| 2 | 0 | 45 | 90 | 135 |
| 3 | 0 | 90 | 180 | 270 |
| 4 | 0 | 135 | 270 | 45 |
| 5 | 0 | 180 | 0 | 180 |
| 6 | 0 | 225 | 90 | 315 |
| 7 | 0 | 270 | 180 | 90 |
| 8 | 0 | 315 | 270 | 225 |

For a detection point $P(r, \theta, \varnothing)$ in the far field, the electric field $E(r)$ can be found from [15,35]. Based on the electric field $E(r)$, the array factor can be found in [16,35], as shown by:

$$f(\theta, \phi) \approx m_r \cdot e^{il\phi} J_l(k_g a \sin \theta) \tag{2}$$

where $m_r$ is a constant related to $r$, $\varnothing$ is the azimuth angle, $k_g$ is the wave vector in water, $l$ indicates the OAM mode, called the "topological charge", $J_l$ represents the $l_{th}$ order Bessel function of the first kind, $a$ is the radius of the array and $\theta$ is the angle between $k_g$ and the propagation direction $z$. The array factor $f(\theta, \varnothing)$ not only can describe the distribution of normalized intensity patterns and phase patterns of the different OAM modes, but can also be utilized to replace the electric field $E(r)$ when calculating the purity of OAM waves. Moreover, it is easy and efficient due to its simple form.

The intensities and the phase distributions of the different generated OAM waves can be obtained from the far field distribution, as depicted in Figure 5. The selected plane has the scale of 40.4 mm × 40.4 mm at $z = 200$ mm, and the minimum value of $\cos \theta$ is 0.9899, from which the propagation distance $z$ can be approximated as $r$. Therefore, the array factor can be read from the CST on one plane with the determined value of $z$. From Figure 5, it can be seen that the phase reference point rotates as $\varnothing$ increases, and the values of the phase change can match the orders of the OAM wave in one turn very well. The intensity distributions of the four modes fit the corresponding Bessel function intensity distributions. Both the phase and intensity distributions demonstrate the super performance of the designed rectangular horn antenna array.

In order to further prove the simulated intensity and phase distributions of generated OAM-carrying waves, we also carried out the simulations by HFSS software, and the corresponding results were obtained and are displayed in Figure 6. It can be seen that these simulated results are consistent with that of the CST, which further verifies the effectiveness of our designs and also demonstrates the correctness of the simulation results.

Meanwhile, in order to show the purity of OAM waves in Figure 5, the mode decomposition is carried out by means of Fourier transform of the field distribution on a circle corresponding to the magnitude maximum [36], as shown in Figure 7a–d, where $m$ is from −3 to 3. Assuming N points on the same circle are selected to calculate the purity of OAM waves, the azimuth spectrum $w_m$ could be obtained and expressed as follows:

$$w_m = \frac{\sum_{n=0}^{N-1} E_n \cdot e^{-im\phi_n}}{\sum_{n=0}^{N-1} \sum_{m=-3}^{3} E_n \cdot e^{-im\phi_n}} \tag{3}$$

where $E_n$ is the electric field of the $n^{th}$ point, $\varnothing_n$ is the theoretical azimuth of the standard vortex field at the location of the $n^{th}$ point.

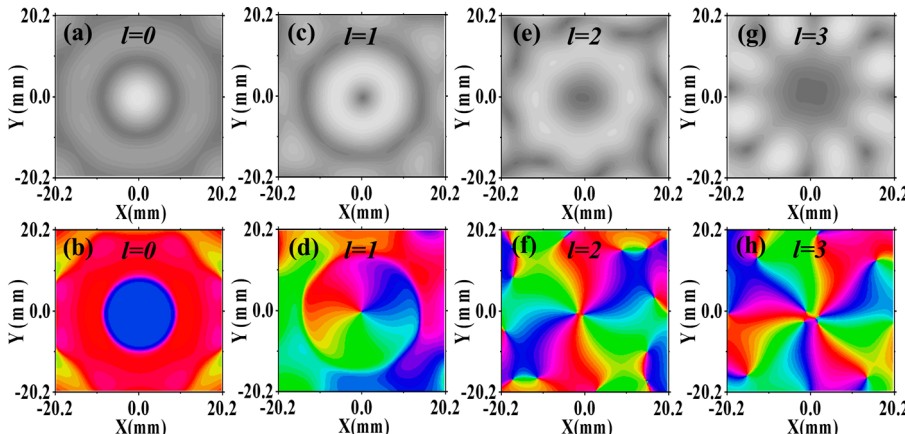

**Figure 5.** Normalized intensity and phase distributions of generated OAM-carrying waves. (**a**) and (**b**) *l* = 0. (**c**) and (**d**) *l* = 1. (**e**) and (**f**) *l* = 2. (**g**) and (**h**) *l* = 3. The horizontal axis is X∈ (−20.2 mm, 20.2 mm), while the vertical axis is Y∈ (−20.2 mm, 20.2 mm).

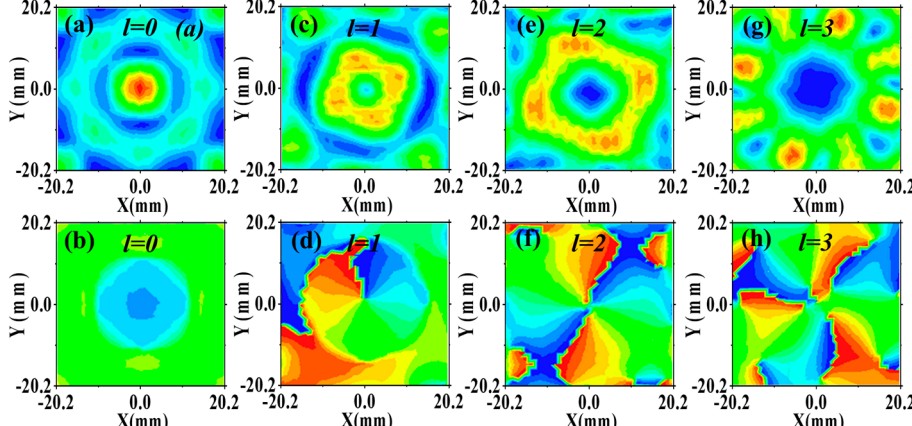

**Figure 6.** Normalized intensity and phase distributions of generated OAM-carrying waves by ANSYS HFSS. (**a**) and (**b**) *l* = 0. (**c**) and (**d**) *l* = 1. (**e**) and (**f**) *l* = 2. (**g**) and (**h**) *l* = 3. The horizontal axis is X∈ (−20.2 mm, 20.2 mm), while the vertical axis is Y∈ (−20.2 mm, 20.2 mm).

The generated OAM mode of $m$ = 1 shows a high fidelity to the design, as shown in Figure 5c,d. The mode decomposition result is demonstrated in Figure 6b, and the energy in the OAM mode of $m$ = 1 clearly dominates, with a proportion above 90%, which is 30 times above the secondary mode, showing a good performance of the antenna array. The remaining OAM modes (OAM mode $m$ = 0 in Figure 5a,b, $m$ = 2 in Figure 5e,f and $m$ = 3 in Figure 5g,h) are similar to the OAM mode of $m$ = 1, as shown in Figure 6a,c,d, respectively.

Figure 7e–h also show the field intensity distributions (along the radial direction) $w_m$ of the different OAM modes at 2.6 GHz, where the points of $\varnothing = 0$ were selected to calculate $w_m$, and $w_m$ equals array factor $f$ in theory. Therefore, the simulation results were compared with the results from numerical calculations to make sure they are consistent. It can be seen that the lines match the center area very well. However, due to the edge effect of the OAM wave, there are small differences at the edge. This phenomenon can be attributed to the increasing difference between $z$ and $r$ with increasing $\theta$, and the difference between two lines will also increase in theory, but it can be accepted totally.

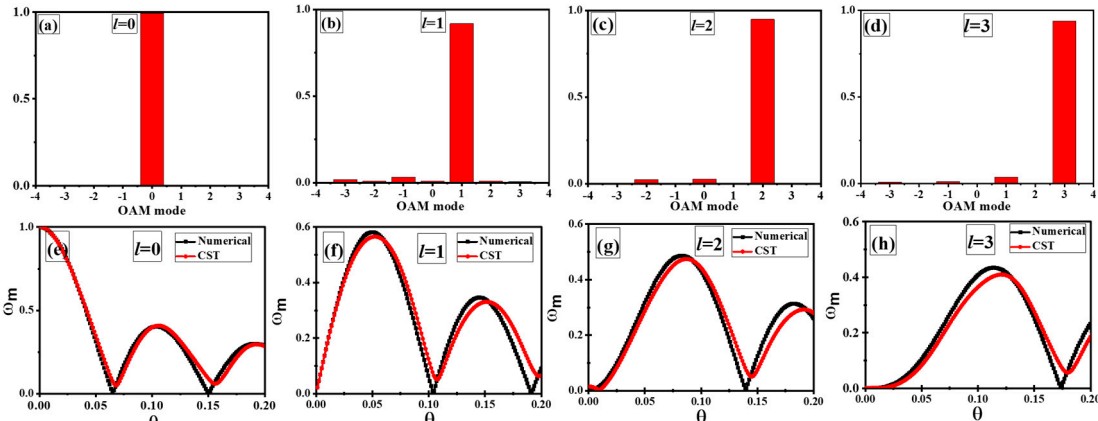

**Figure 7.** Mode decomposition of the different OAM modes at 2.6 GHz. (**a**) *l* = 0, (**b**) *l* = 1, (**c**) *l* = 2 and (**d**) *l* = 3. Field intensity distributions (along the radial direction) of the CST simulation and numerical calculation at 2.6 GHz. (**e**) *l* = 0, (**f**) *l* = 1, (**g**) *l* = 2 and (**h**) *l* = 3.

## 4. Conclusions

In this paper, the experiment of the dielectric-loaded rectangular horn antenna was investigated, which works at 2.6 GHz in water. The experimental results show that this antenna has good impedance characteristics ($S_{11} < -10$ dB) and reasonable losses (less than 3.5 dB total for two antennas and the coaxial line) from 2.5 GHz to 2.7 GHz. A water-immersed OAM rectangular horn antenna array, working at 2.6 GHz, was also proposed to realize the OAM wave radiation in a water environment based on the above single antenna. From the obtained phase and intensity distributions of the generated OAM waves with different modes, it can be indicated that the designed antenna array works very well at 2.6 GHz, which was also further proved by analyzing the purity of the generated OAM waves. The OAM waves can radiate far enough to meet the requirements for future underwater applications of OAM, such as underwater communication and underwater imaging for biological tissues and vegetable rhizomes. In future, we will try to generate wideband underwater OAM waves and to apply the generated OAM waves to the concrete underwater applications.

**Author Contributions:** Conceptualization, Y.Y. and S.W.; methodology, Y.Y. and Z.W. (Zhe Wu); software, Y.Y. and H.J.; formal analysis, F.S. and B.Z.; experiment, Y.Y. and Z.W. (Zhanliang Wang), writing—original draft preparation, Y.Y.; writing—review and editing, Z.G. and Q.Z.; supervision, Y.G.; funding acquisition, Z.G. and Y.G.

**Funding:** This research received no external funding.

**Acknowledgments:** This work was supported by National Natural Science Foundation of China (NSFC) (Grant Nos. 61531010, 61775050); Fundamental Research Funds for the Central Universities (PA2019GDZC0098).

**Conflicts of Interest:** The authors declare no conflict of interest.

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
