# Peer review of "Designing a Water-Immersed Rectangular Horn Antenna for Generating Underwater OAM Waves"

_electronics, doi:10.3390/electronics8111224_

Round 1

Reviewer 1 Report

My comments now have been addressed. 

Author Response

Great Thanks!

Reviewer 2 Report

The authors must explain what they have in mind in the following
statement

"The selected plane has the scale of 40.4mm×40.4mm at z=200mm where the minimum value of cos ? is 0.9899 for keeping that the propagation distance z can be approximately as ? . Therefore, the array factor can be read from the CST on one plane with the determined value of z...."

Reviewer 3 Report

In the Chapter 1 (Introduction), The Authors present a critical literature review on the vortex waves carrying orbital angular momentum (OAM). However, there are some shortcomings and errors that need to be corrected/eliminated.

Substantive remarks

In spite of the water environment, it has not been shown that measurements at such a short distance are taken in a far-field, where we have the TEM wave, where the E and H components are mutually perpendicular (the Poynting vector S=E×H is real and oriented radially). This problem is not solved by the Authors’ declaration that: “(...) due to the large loss of water and the enough space of the tank, noises and most reflected waves can be absorbed, so it can be regarded as an infinite space that is also called anechoic environment”. This is also not confirmed by the assumption made on the basis of the Fig. 4(d). Beside above, in the Fig. 3(d) the measured radiation pattern is shown. There is no comparison of measurement results and numerical calculations form the CST software. In the opinion of the reviewer, the proposed title should be corrected for two words concerning the same phenomenon (“water-immersed”, “underwater”).

Editorial corrections:

all symbols used in text/equations/figures/tables should be formatted uniformly (italics for scalar variables in the whole manuscript); in this context, please see the Guide for Authors; with some exceptions such as “%”, the unit should be separated from the value by a space (please use 50 Ω instead of 50Ω, 2.7 GHz instead of 2.7GHz, 40 cm instead of 40cm, etc.) and it should be written in a single line of text (e.g. 2.6 GHz); please use “radiation pattern” instead of “radiation characteristic”; please eliminate any remaining errors.

Reviewer 4 Report

The paper presents a rectangular horn designed for generating underwater OAM waves. It is well written and the conclusions are supported by experimental results. The possible applications of the proposal are also explained in the text.

Only a small spelling in line 36, "has a rotating" probably should be "have a rotating".

For a better comparison, the distances and losses of other underwater reported should be mentioned in the text.

Round 2

Reviewer 3 Report

The presented explanations are sufficient. The reviewer thanks the Authors for their corrections and answers. Finally, please eliminate minor editorial errors:

all symbols used in text/equations/figures/tables should be formatted uniformly (italics for scalar variables in the whole manuscript); in this context, please see the Guide for Authors; please eliminate any remaining errors.

This manuscript is a resubmission of an earlier submission. The following is a list of the peer review reports and author responses from that submission.

Round 1

Reviewer 1 Report

Although authors had addressed most of the comments, but the simulated results for the antenna array model didn't cross-check by authors. So, my concern is still remained.

Authors said that only one software can model their proposed work. But, i dont think this is valid statement. Software such as Comsol multiphysics, Ansys HFSS, XFDTD and others. Surely, one can be used to verify their model. 

Reviewer 2 Report

I would like to thank the Authors for considering my comments. Figure 3c seems ok.

I do not understand why the additional simulations cannot be included in the paper "due to the limits in the manuscript". Electronics, as far as I know, has no page limits.

Moreover, it is still not fully demonstrated why the proposed design is better than an open waveguide for the generation of underwater OAM waves. For example, it will be interesting to see how does Figure 6 change if an open waveguide is used instead of the proposed one.

Reviewer 3 Report

This article aims to design and test a single water-immersed rectangular horn antenna, which can generate EM waves underwater. By introducing the dielectric-loaded waveguide in this antenna, the sealing problem of the antenna can be solved while the established position and length of the feed probe can be ensured, so the frequency offset can be reduced to a certain extent. The experimental results of single antenna are consistent with the simulation ones. Based on the above single antenna, the water-immersed OAM rectangular horn antenna array is also designed and simulated. Through simulations, the high-fidelity vortex fields with the OAM modes of l = 0, 1, 2, 3 can be generated efficiently in far field, and meanwhile, the purity of these OAM modes have been obtained from the vortex field distributions.

Overall, the paper presents a reasonable scientific contribution. The paper is now well-structured, and the ideas are properly presented. Moreover, the suggested revisions have been properly incorporated.

Round 2

Reviewer 1 Report

Authors didn't show how the given comments/responses being implemented within the revised manuscript. It is very important to prove the results in the paper has been fully validated. 

Therefore, my decision remains as major revision.